# Social Elite in Imperial China: Their Destinies as Documented by the Historical Literature and Their Personality as Defined by the Contemporary Five-Factor Model

**Bingren Zhang** [1], **Hongying Fan** [2] and **Wei Wang** [3,*]

1   School of Clinical Medicine, Hangzhou Normal University, Hangzhou 311121, China
2   College of Psychology, Beijing Sport University, Beijing 100084, China
3   Department of Psychology, Norwegian University of Science and Technology, 7491 Trondheim, Norway
*   Correspondence: wew@ntnu.no; Tel.: +47-46542194

**Abstract: Background**: The association between personality and life outcome has been widely studied in Western countries, and one might question whether the association exists in China. The official documentation from the *Twenty-Six Histories* of Imperial China, which presents life-long data on the social elite, may offer a convenient way to realize this effort. Meanwhile, a possible association might help identify competent personalities and offer treatment hints for personality disorders or other psychiatric deviations worldwide. **Methods**: Based on these historical records (about 618–1911 AD) on 18 social elite groups with long longevity (Macrobian group) and 30 with normal lifespans (Control group), we assessed personality traits/facets using the revised NEO Personality Inventory (NEO-PI-R) and destiny using the Destiny Evaluation Questionnaire (DEQ). **Results**: Compared to the Controls, the Macrobian group scored higher on the DEQ's Health and Destiny in General and lower on the NEO-PI-R traits Openness to Experience and Extraversion and facets such as Openness to Fantasy, Openness to Aesthetics, Openness to Feelings, Excitement-Seeking, and Self-Consciousness. In the Macrobian group, the Trust and Compliance facets predicted the DEQ's Family and Marriage and Social Relationships aspects, respectively; Conscientiousness and its facets Dutifulness, Self-Discipline, and Competence predicted Family and Marriage, Career Achievement, and Destiny in General, respectively; and the Self-Consciousness facet predicted worse performance in Career Achievement, Family and Marriage, and Social Relationships and the Depression facet of Destiny in General. In the Control group, Openness to Feelings positively and Anxiety negatively predicted Health. **Conclusions**: Less self-focused attention and more interdependence between individuals were beneficial to several aspects of individual destiny in Imperial China, which might be profound for the individual career development and clinical treatment of personality disorders in contemporary society.

**Keywords:** destiny description; five-factor model of personality; Imperial China; lifespan; social elite

## 1. Introduction

Lifespan development, also called destiny, is a chronologically organized course in human lives. An individual's destiny is not only subject to the era and sociocultural environment of his/her time but also to personal factors. Personality, for instance, plays an important role in occupational performance, interpersonal relationships, and other life functions [1]. It has been defined as the enduring characteristics or disposition of an individual that differentiates him/her from the standard of a normal person with a similar social background [2]. To be more specific, a personality with high integration, which is congruent with organismic needs, leads to health and wellbeing [3]. A disordered personality, on the other hand, leads to distress or impairment and predicts the course of disease and therapeutic effects of mental disorders [4,5].

A highly integrated personality is characterized by high levels of extraversion, agreeableness, conscientiousness, and openness and low levels of neuroticism (or high levels of emotion

stability) [6]. Using the five-factor model (or Big Five) [7,8], many studies have shown the relations between personality traits and specific life outcomes. For example, Conscientiousness was modestly associated with overall job performance across occupations [9,10]. Conscientiousness or social dependability contributed to health status and longevity [11,12]. Moreover, a high level of General Activity (a facet of Extraversion) was found to lead to a long lifespan [13]. Regarding family relationships, high levels of Agreeableness and Emotion Stability were the best predictors of marital satisfaction [14]; Openness to Experience and Extraversion were related to a more nurturing and less restrictive parenting style [15]; and Agreeableness played an important role in maintaining satisfactory parenting styles [16]. Regarding an individual's social life, high levels of Extraversion and Openness improved social status and reduced social anxiety, respectively [17,18]. In addition, high Neuroticism (or low Emotion Stability) was detrimental to many of the life domains listed above [13,16,17]. Recent work has revealed other fundamental factors in performance and achievement, namely passion, grit, and growth mindset [19,20]. The trait grit helps passionate individuals to develop their long-term goals [19,21]. Grit is associated with conscientiousness [22], motion, growth mindset, and lifelong brain structure development [23,24].

So far, investigations of this kind have mainly been carried out in Western countries where individualism and self-reliance are more emphasized rather than in Oriental countries where collectivism and harmony are valued [25,26]. Interestingly, individualism and self-reliance were associated with achieving individual success [27,28] but less beneficial to health and longevity, which required high social dependability or conscientiousness [11]. On the contrary, a collectivist culture might provide enough social dependability [25]. Moreover, Chinese people rated their first personality trait as Intelligent (or intelligence, which is similar to Openness to Experience) [29], which was essential for individuals to attain success or contribute to society [30]. In addition, there were mutual relationships between social accomplishment and personality mobilization through the social values of abiding, personal effort, and family support in Chinese culture [31,32]. Further, social factors often act as an important element for psychological wellbeing in China, especially in the Imperial period [33]. Thus, there might be different aspects/levels of personality that contribute to life outcomes, especially individual achievement, health, and longevity, in Oriental culture, at least among the Chinese. Furthermore, most previous studies used broad personality factors (traits); however, more detailed facets might be beneficial to increase the consistency and validity of life outcome predictions. Unfortunately, except for a few longitudinal studies [12,13], the available results on personality's influence on destiny factors are not lifelong, and most end at a specific stage of the life of an individual.

In the current study, we explore the lifelong data of personality descriptions and career or personal development in a group of social elites rather than conducting an in-depth study on one single individual. These individuals had great social influence, and their personal lives, endeavors, and achievements were documented in relative detail in the official historical records, which might offer a convenient way to debrief personal characteristics and lifelong developments. This investigation might also offer suggestions for competent personality development and the clinical management of personality disorders or other psychiatric alterations worldwide. We would like to use classical documentation that bears the lifelong records of individuals who were deeply influenced by aspects of Chinese culture such as collectivism and harmony [26] to explore the detailed associations between personality traits/facets and life outcomes over the progression and development of Chinese history. Individuals were classified into the Macrobian group and the Control group with seventy years old as the cut-off point of their lifespan, as according to the ancient Chinese sage Confucius, "humans rarely live to 70". The personality traits/facets were assessed with the revised NEO Personality Inventory (NEO-PI-R) [7], a worldwide-tested questionnaire assessing a five-factor model of personality. The Destiny Evaluation Questionnaire (DEQ) was developed by the authors to assess factors of destiny. Based on previous results, we hypothesized that in Imperial China: (1) the Macrobian group would be more socially dependent and score higher on Conscientiousness and its facets than their

counterparts did; and (2) Openness to Experience and its facets, as an important part of Chinese personality, would be associated with the destiny structures of social elites.

## 2. Methods

### 2.1. Characters Studied

The documentation source was the *Twenty-Six Histories* of China, i.e., the *Twenty-Four Histories* of China [34] plus the *New History of the Yuan Dynasty* [35] and the *Draft History of Qing* [36], a series of records of the dynastic histories from remote antiquity until the Qing Dynasty written in a unified biographical form. Forty-eight social elites that lived in Imperial China, mainly between two periods of great prosperity, the Tang and Qing dynasties (618–1911 AD), were selected (Table 1). They scored 4 or more on the 7-point Likert scale for the Career Achievement domain of the Destiny Evaluation Questionnaire (DEQ, see below Section 2.2.2). Eighteen characters were categorized into the Macrobian group (17 men and one woman; mean age: 78.28 ± 5.36; range: 70–89), and 30 were categorized into the Control group (all men; mean age: 55.63 ± 8.34; range: 39–66). The group difference in lifespan was statistically significant (t = 11.45, *df* = 46, *p* < 0.001). The characters in each group were further classified by ethnicity, dynasty, family background, education level, and career (Table 1). No significant difference was found between the Macrobian and Control groups regarding gender (Mann–Whitney U = 255.00, *p* = 0.38), ethnicity (U = 258.00, *p* = 1.00), dynasty (U = 257.50, *p* = 0.77), family socioeconomic status (U = 265.50, *p* = 0.94), education level (U = 269.00, *p* = 1.00), or career (U = 257.50, *p* = 0.74). No ethics issue was involved in this study.

**Table 1.** The distributions of characters in Macrobian (*n* = 18) and Control (*n* = 30) groups, and their ethnicity, dynasty, family socioeconomic status, education level, and career.

| | Macrobian | Controls |
|---|---|---|
| Character | He TANG ( 汤和, Ming Dynasty), Shanchang LI ( 李善长, Ming), Yuan HUI ( 慧远, Jin Dynasty), Zhe SU ( 苏辙, Song Dynasty), Zetian WU ( 武则天, Tang Dynasty), Guancheng FANG ( 方观承, Qing Dynasty), Bao LE ( 勒保, Qing), Tang QIAN ( 钱唐, Ming), Li JIAO ( 焦礼, Ming), Lun DONG ( 董伦, Ming), Zhi YI ( 仪智, Ming), Ying WANG ( 王英, Ming), Xili QIAN ( 钱习礼, Ming), Jin GUO ( 郭璡, Ming), Zhongfu LIU ( 刘中敷, Ming), Xuan ZHOU ( 周瑄, Ming), Ding YANG ( 杨鼎, Ming), Sanwu LIU ( 刘三吾, Ming) | Wenzhong LI ( 李文忠, Ming), Yu DENG ( 邓愈, Ming), Ying MU ( 沐英, Ming), Qian YU ( 于谦, Ming), Guofan ZENG ( 曾国藩, Qing), Xun SU ( 苏洵, Song), Yangming WANG ( 王阳明, Ming), Anshi WANG ( 王安石, Song), Rong Chai ( 柴荣, Five Dynasties and Ten Kingdoms), Yu LI ( 李煜, Five Dynasties and Ten Kingdoms), Kuo SHEN ( 沈括, Song), Zicheng LI ( 李自成, Ming), Zhongxian WEI ( 魏忠贤, Ming), Sangui WU ( 吴三桂, Qing), He ZHENG ( 郑和, Ming), Xueyan HU ( 胡雪岩, Qing), Zhao ZHANG ( 张照, Qing), Yike HAN ( 韩宜可, Ming), Shilu LI (李仕鲁, Ming), Yong HE ( 和勇, Ming), Qian LIANG ( 梁潜, Ming), Ji CHEN ( 陈济, Ming), Xu ZHOU ( 周叙, Ming), Ben ZHANG ( 张本, Ming), Dun GUO ( 郭敦, Ming), Feng ZHANG ( 张凤, Ming), E LIN (林鹗, Ming), Yanliang GUI ( 桂彦良, Yuan Dynasty), Jurun ZAN ( 昝居润, Five Dynasties and Ten Kingdoms), Gu LI ( 李穀, Five Dynasties and Ten Kingdoms) |
| Ethnicity (Han/others) | 16/2 | 28/2 |
| Dynasty (Qing/Ming/Yuan/Song/ Five Dynasties and Ten Kingdoms/ Tang/the Northern and Southern Dynasties/Jin Dynasty) | 2/13/0/1/0/1/0/1 | 4/18/1/3/4/0/0/0 |
| Family socioeconomic status (merchant/artisan/farmer/scholar/royal) | 0/0/11/7/0 | 2/0/16/11/1 |
| Education level (none/primary/secondary/higher education) | 3/3/2/10 | 3/4/9/14 |
| Career (politics/science culture/economy) | 17/1/0 | 27/2/1 |

*2.2. Materials and Measures*

2.2.1. The Revised NEO Personality Inventory (NEO-PI-R)

The NEO-PI-R [37] is a 240-item scale. It contains five factors, namely Neuroticism, Extraversion, Openness to Experience, Agreeableness, and Conscientiousness, which are further classified into 30 facets, with six facets for each factor and eight items for each facet. The NEO-PI-R has been shown to be reliable and valid not only in Western but also in Asian cultures [38,39]. Given that the descriptions of specific personality traits in historical documentation are often limited, we used a forced-choice rating (yes = 1; no = 0; for judge's reasoning, see Section 2.3.) to standardize the results. In the current study, the Cronbach's alpha scores for NEO-PI-R factors ranged from 0.74 to 0.94 in the Macrobian group and 0.82 to 0.92 in the Control group.

2.2.2. The Destiny Evaluation Questionnaire (DEQ)

The DEQ (authors' work, see the Supplementary Material for details) assesses the lifespan development of an individual using a 7-point Likert scale. It contains four domains, each consisting of 3 to 16 statements. The first domain, Career Achievement, assesses occupational attainment in terms of number of works, social influence, reputation in later generations, etc. The individuals in the fields of politics, military, technology, culture, and economics were assessed according to their achievements. The second domain, Health, evaluates physical condition based on the cause of death, illness severity, and length of lifespan. The third domain, Family and Marriage, contains statements on relationships with family members, his/her evaluation by relatives, marital condition, etc. The fourth domain, Social Relationships, assesses network breadth, evaluation by close friends or opponents, and number of friends. The mean subscores of the Career Achievement and Family and Marriage domains and the overall scores of the Health and Social Relationships domains were summed and used as the total score of Destiny in General. The Cronbach's alpha scores for DEQ as a whole were 0.83 in the Macrobian group and 0.75 in the Control group, and those of the DEQ Career Achievement and Family and Marriage domains were 0.85 and 0.75, respectively, in the Macrobian group and 0.75 and 0.74, respectively, in the Control group.

*2.3. Scoring Procedure*

Six judges (three MSc candidates in psychology; one MD candidate in psychiatry; and two PhD holders in psychiatry, one of whom had a background in classical Chinese documentation) voted on each item of the NEO-PI-R (0 or 1) based on the records in the *Twenty-Six Histories* of China. Each item was voted on by six judges independently and labeled as a meaningful score if it received more than three "yes" votes. If an item received three "yes" and three "no" votes, the seventh judge (the corresponding author, a DSc holder) made the final decision. For the DEQ however, the judges rated each item first, and then their mean scores on each item were taken for further analyses.

*2.4. Statistical Analyses*

In the two groups, the age distribution as well as scores on the NEO-PI-R and DEQ scales were compared using Student's *t* test. The gender, ethnicity, dynasty, family socioeconomic status, education level, and career distributions in the two groups were submitted to the Mann–Whitney U test. The effect sizes of the significant group differences were calculated using Cohen's d. We also applied a multiple linear regression analysis (stepwise method) to explore the predictions of the NEO-PI-R factors/facets for the DEQ domains, taking ethnicity, dynasty, family socioeconomic status, education level, and career as covariates. The *p* values for group comparisons were set at 0.05. In order to reduce the risk of a Type I error, we considered $p < 0.05$, beta > 0.30, and adjusted $R^2 > 0.25$ as significant regarding predictions.

## 3. Results

### 3.1. Scores of Personality and Destiny (Hypothesis 1)

The Macrobian group scored significantly lower than the Control group on the NEO-PI-R Openness to Experience (t = −2.27, *df* = 46, *p* = 0.03, Cohen's d = 0.63) and Extraversion (t = −2.46, *df* = 46, *p* = 0.02, Cohen's d = 0.55) traits. The results on the facet level showed that the Macrobian group scored lower on Openness to Fantasy (t = −3.23, *df* = 35, *p* < 0.01, Cohen's d = 0.62), Openness to Aesthetics (t = −2.33, *df* = 46, *p* = 0.02, Cohen's d = 0.56), Openness to Feelings (t = −2.48, *df* = 46, *p* = 0.02, Cohen's d = 0.59), Excitement-Seeking (t = −2.91, *df* = 40, *p* < 0.01, Cohen's d = 0.59), and Self-Consciousness (t = −2.64, *df* = 37, *p* = 0.01, Cohen's d = 0.52). No significant differences were found for other factors such as Neuroticism (t = −1.91, *df* = 43, *p* = 0.06), Agreeableness (t = 0.40, *df* = 46, *p* = 0.69), or Conscientiousness (t = −1.01, *df* = 46, *p* = 0.32) or other facets (ps = 0.06–0.91). In the DEQ, the Macrobian group scored significantly higher than the Control group in Health (t = 4.73, *df* = 46, *p* < 0.01, Cohen's d = 1.30) and Destiny in General (t = 2.93, *df* = 46, *p* < 0.01, Cohen's d = 0.89). No significant differences were found for other domains such as Career Achievement (t = −1.06, *df* = 46, *p* = 0.30), Family and Marriage (t = −0.18, *df* = 46, *p* = 0.86), and Social Relationships (t = 1.21, *df* = 46, *p* = 0.23) (Table 2).

**Table 2.** Mean scores (±S.D.) on the Revised NEO Personality Inventory factors and facets as well as the Destiny Evaluation Questionnaire in Macrobian (*n* = 18) and Controls (*n* = 30).

| | Macrobian | Controls |
|---|---|---|
| Revised NEO Personality Inventory | | |
| Neuroticism | 8.06 ± 3.36 | 11.09 ± 7.57 |
| N1-Anxiety | 0.96 ± 1.08 | 1.72 ± 2.06 |
| N2-Angry Hostility | 1.39 ± 1.17 | 1.98 ± 2.17 |
| N3-Depression | 0.15 ± 0.31 | 0.64 ± 1.45 |
| N4-Self-Consciousness | 0.70 ± 0.57 * | 1.69 ± 1.90 |
| N5-Impulsiveness | 2.13 ± 0.89 | 2.47 ± 1.47 |
| N6-Vulnerability | 2.72 ± 1.91 | 2.59 ± 1.85 |
| Extraversion | 16.85 ± 4.78 * | 21.56 ± 8.48 |
| E1-Warmth | 2.20 ± 1.17 | 2.90 ± 2.06 |
| E2-Gregariousness | 3.87 ± 1.12 | 4.51 ± 1.13 |
| E3-Assertiveness | 4.61 ± 1.77 | 4.99 ± 1.64 |
| E4-Activity | 3.44 ± 1.60 | 4.14 ± 1.58 |
| E5-Excitement-Seeking | 0.74 ± 0.96 ** | 2.30 ± 2.66 |
| E6-Positive Emotions | 1.98 ± 0.83 | 2.71 ± 1.73 |
| Openness to Experience | 19.74 ± 5.00 * | 23.66 ± 6.20 |
| O1-Openness to Fantasy | 1.28 ± 0.42 ** | 2.26 ± 1.57 |
| O2-Openness to Aesthetics | 2.09 ± 1.36 * | 3.23 ± 2.02 |
| O3-Openness to Feelings | 2.41 ± 1.38 * | 3.67 ± 2.14 |
| O4-Openness to Actions | 4.39 ± 0.77 | 4.26 ± 0.82 |
| O5-Openness to Ideas | 4.80 ± 1.55 | 5.33 ± 1.48 |
| O6-Openness to Values | 4.78 ± 1.50 | 4.91 ± 0.97 |
| Agreeableness | 26.06 ± 5.13 | 25.26 ± 7.58 |
| A1-Trust | 3.50 ± 1.34 | 3.63 ± 1.62 |
| A2-Straightforwardness | 6.52 ± 1.59 | 6.06 ± 2.03 |
| A3-Altruism | 3.50 ± 1.17 | 3.72 ± 1.60 |
| A4-Compliance | 4.50 ± 1.16 | 4.08 ± 1.71 |
| A5-Modesty | 5.65 ± 1.11 | 4.97 ± 1.71 |
| A6-Tender-Mindedness | 2.39 ± 1.42 | 2.80 ± 2.12 |

**Table 2.** *Cont.*

|  | Macrobian | Controls |
|---|---|---|
| Conscientiousness | 27.50 ± 7.92 | 29.74 ± 7.15 |
| C1-Competence | 4.41 ± 1.51 | 4.73 ± 1.49 |
| C2-Order | 3.46 ± 1.33 | 3.90 ± 1.55 |
| C3-Dutifulness | 4.39 ± 1.79 | 4.94 ± 1.63 |
| C4-Achievement Striving | 5.20 ± 2.10 | 6.20 ± 1.31 |
| C5-Self-Discipline | 5.15 ± 1.69 | 5.20 ± 1.55 |
| C6-Deliberation | 4.89 ± 1.15 | 4.77 ± 1.49 |
| Destiny Evaluation Questionnaire |  |  |
| Career Achievement | 5.31 ± 0.59 | 5.48 ± 0.52 |
| Health | 6.45 ± 0.94 ** | 4.89 ± 1.20 |
| Family and Marriage | 4.97 ± 0.67 | 5.01 ± 0.78 |
| Social Relationship | 5.48 ± 0.68 | 5.18 ± 0.93 |
| Destiny in General | 22.21 ± 1.92 ** | 20.59 ± 1.83 |

Note: *, $p < 0.05$ vs. Controls; **, $p < 0.01$ vs. Controls.

### 3.2. Associations between Personality and Destiny (Hypothesis 2)

When predicting the DEQ using the NEO-PI-R, the accounted variances (adjusted $R^2$s) ranged from 0.41 to 0.87 in the Macrobian group and from 0.13 to 0.40 in the Control group. At the factor level, in the Macrobian group, the NEO-PI-R Conscientiousness positively and dynasty and family socioeconomic status negatively predicted the DEQ Family and Marriage. At the facet level, in the Macrobian group, Self-Discipline positively and Self-Consciousness negatively predicted Career Achievement; Dutifulness and Trust positively and dynasty and Self-Consciousness negatively predicted Family and Marriage; Compliance positively and Self-Consciousness negatively predicted Social Relationships; and Competence positively and Depression negatively predicted Destiny in General. In the Control group, Openness to Feelings positively and Anxiety negatively predicted Health (Table 3).

**Table 3.** Predictions of Destiny Evaluation Questionnaire by Revised NEO Personality Inventory from factor and facet levels (ethnicity, dynasty, career, family socioeconomic status, and education as covariates) using Stepwise Regression Analysis in Macrobian ($n$ = 18) and Controls ($n$ = 30).

|  | a-$R^2$ | Macrobian<br>Beta (B, SE), Predictor | a-$R^2$ | Controls<br>Beta (B, SE), Predictor |
|---|---|---|---|---|
| Career Achievement | 0.41 | −0.44 (−0.45, 0.20), N4-Self-Consciousness<br>0.44 (0.15, 0.07), C5-Self-Discipline |  |  |
| Health |  |  | 0.30 | −0.68 (−0.40, 0.11), N1-Anxiety<br>0.55 (0.31, 0.11), O3-Openness to Feelings |
| Family and Marriage | **0.76**<br>0.87 | **−0.81 (−0.31, 0.05), dynasty**<br>**0.68 (0.06, 0.01), Conscientiousness**<br>**−0.37 (−0.50, 0.21), family socioeconomic status**<br>−0.67 (−0.26, 0.04), dynasty<br>0.46 (0.17, 0.04), C3-Dutifulness<br>−0.54 (−0.64, 0.15), N4-Self-Consciousness<br>0.36 (0.18, 0.06), A1-Trust |  |  |
| Social Relationship | 0.46 | −0.58 (−0.68, 0.21), N4-Self-Consciousness<br>0.39 (0.23, 0.11), A4-Compliance |  |  |
| Destiny in General | 0.54 | −0.67 (−4.17, 1.04), N3-Depression<br>0.47 (0.59, 0.21), C1-Competence |  |  |

Note: a-$R^2$, adjusted $R^2$; only predictors having $p < 0.05$ and beta > 0.30 under the prediction effect of adjusted $R^2 > 0.25$ were listed; predictions analyzed from factor level were bolded; for facet name codes, see Table 2.

## 4. Discussion

To the best of our knowledge, this is the first study exploring the relationship between personality and destiny in a group of social elites in Imperial China. We found that compared to their counterparts, the Macrobian group had higher scores in Health and Destiny in General and lower scores in Openness to Experience and Extraversion, particularly the facets of Openness to Fantasy, Openness to Aesthetics, Openness to Feelings, Excitement-Seeking, and Self-Consciousness. In the Macrobian group, Trust and Compliance predicted interpersonal relationships; Conscientiousness and its facets Dutifulness, Self-Discipline, and Competence predicted Family and Marriage, Career Achievement, and Destiny in General, respectively; the Self-Consciousness facet negatively predicted Career Achievement, Family and Marriage, and Social Relationships; and the Depression facet negatively predicted Destiny in General. Therefore, our first hypothesis regarding the influence of social dependence or Conscientiousness on longevity was not confirmed; nonetheless, our second hypothesis regarding destiny and Openness to Experience and its facets was partially supported.

In the Control group, the negative prediction of the Anxiety trait in the Health domain was in line with the fact that neurotic patterns such as chronic anxiety lead to sorts of health problems and social dysfunctions [40]. The positive prediction of Openness to Feelings for Health might be explained by the fact that openness, especially to feelings, predicted health concerns and health-related search behavior [41], which might promote health management and lead to an elevated health level.

In the Macrobian group, higher Health and Destiny in General were no surprise because health or wellbeing leads to a longer lifespan. Moreover, the Macrobian group displayed lower Openness to Experience levels, especially the Openness to Fantasy, Openness to Aesthetics, and Openness to Feelings facets, instead of higher Conscientiousness as we hypothesized. These results were inconsistent with some reports in Western countries that demonstrated a link between openness traits and health and longevity [42,43], suggesting that one key to a long lifespan under the Chinese collectivist culture was to reduce one's indulgence in the unshared part of the self or the pursuit of personal feelings and experiences. Similarly, we found lower levels of Extraversion, especially Excitement-Seeking, in the Macrobian group, which also supported the above viewpoint. Furthermore, Self-Consciousness, the stable tendency to focus attention on oneself, was lower in the Macrobian group and predicted worse performance in many life outcomes such as Career Achievement, Family and Marriage, and Social Relationships, which generally agrees with previous results showing that Neuroticism negatively affected occupational performance and interpersonal relationships [13,16,17]. Indeed, evidence showed that high levels of Self-Consciousness lead to anxiety, depression, and loneliness and decrease the self-worth of individuals [44]. Additionally, Trust and Compliance predicted family and social relationships, respectively, which is partly supported by the association between agreeableness and family and peer relations [14,45]. These results further evidenced that under the Chinese culture of collectivism and harmony, interpersonal relationships, the interdependence of individuals (the manifestation of Trust), and the inhibition of overt negative emotional expression or aggressive behaviors (the manifestation of Compliance) were encouraged, contributing to the steadiness of the hierarchy in ancient China [25,26].

In the Macrobian group, the predictions of Conscientiousness and its facet Dutifulness for Family and Marriage, Self-Discipline for Career Achievement, and Competence for Destiny in General were supported by the negative influence of low conscientiousness on intergenerational relationships [46] and by the positive associations between Conscientiousness and occupational performance [10], longevity [11], etc. Similarly, both momentary and trait self-control were reported to correlate strongly with happiness and well-being [47,48]. For the accomplishment of long-term goals, recent research has shown that grit, being linked with conscientiousness and perseverance, leads to better performance [19] and to further development of brain structures [24]. In addition, the negative predictions of dynasty (the more remote the dynasty from the present age, the lower the Family and Mar-

riage score) and family socioeconomic status for Family and Marriage might be because the highlighted sense of hierarchy emphasized obedience to the paternal role instead of family harmony [33].

However, one should also bear in mind the limitations of our study design. First, the historical documents were many years old and had limited and sometimes coarse contents. Thus, they cannot provide detailed or ideal psychobiographic descriptions of an individual. However, our endeavor is an easy method of tracing the associations between personality traits and facets relating to the destiny of social elites in Imperial China. Second, due to the male dominance in Feudal/Imperial China, we had fewer women to choose from for the study, and therefore our conclusions need gender-balanced verification. Third, there are few quantitative descriptions in the historical records of health conditions and social relationships, and the scales for these two dimensions were therefore simplified in our DEQ test. Fourth, we used the force-choice (yes vs. no) rating on each NEO-PI-R item, which might characterize personality less accurately than using a 5-point Likert scale. Future studies might be conducted on the general population or to psychiatric patients.

Nevertheless, at the facet level, we found lower openness to experience, excitement-seeking, and self-consciousness in the Macrobian group. We demonstrated the associations between conscientiousness and career achievement, family relationship, and destiny in general; between neuroticism and many aspects of destiny; and between agreeableness and interpersonal relationships in the Macrobian group. We also found an association between openness to feelings and health in the Control group. Therefore, our data on social elites set an example for contemporary individuals of Eastern and Western cultures to prepare their lifelong careers and help shape their personality structures according to different societal needs and offer therapeutic advice for the treatment of psychiatric diseases including personality disorders.

## 5. Conclusions

Our study revealed that less self-indulgence or self-focused attention and more devotion to and interdependence between individuals were beneficial to several aspects of individual destiny in Imperial China, which might offer some hints regarding personality development and psychiatric therapeutics in contemporarily Eastern and Western societies.

**Supplementary Materials:** The following supporting information can be downloaded at: https://www.mdpi.com/article/10.3390/psychiatryint4010006/s1, the English translation and Chinese version of Destiny Evaluation Questionnaire.

**Author Contributions:** W.W. conceived the study, B.Z. and H.F. contributed to the study design and collected the data, B.Z. led the data analysis, B.Z. and W.W. drafted the paper. All authors have read and agreed to the published version of the manuscript.

**Funding:** Bingren Zhang was supported by the start-up research program of Hangzhou Normal University (No. 4125C50220204107).

**Institutional Review Board Statement:** Not applicable.

**Informed Consent Statement:** Not applicable.

**Data Availability Statement:** The datasets used and/or analyzed during the current study are available from the corresponding author on reasonable request.

**Acknowledgments:** We thank Chenyang ZHOU and Liang XU for studying the documented texts, and Jiawei Wang and Xu Shao for data collection.

**Conflicts of Interest:** Regarding research work described in the paper, each one of our co-authors, B.Z., H.F., H.S. and W.W., declares that there is no conflict of interest.

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
