# Peer review of "Social Elite in Imperial China: Their Destinies as Documented by the Historical Literature and Their Personality as Defined by the Contemporary Five-Factor Model"

_2673-5318, doi:10.3390/psychiatryint4010006_

Round 1

Reviewer 1 Report

Authors present here an interesting work exploring the detailed associations between personality traits/facets and life outcomes with the progression and development of in a group of social elite from Imperial Chinese history.
Using revised NEO Personality Inventory (NEO-PI-R), and Destiny Evaluation Questionnaire (DEQ), authors analyzed the personality traits/ facets and factors of destiny between 18 characters, the Macrobian group, and 30 in the counterpart group and found that the less self-indulgence or self-focused attention, and more devotion to and interdependence between individuals were beneficial to several aspects of individual destiny in Imperial China.

Although I appreciate the authors attempt for such an interesting study with well-presented data and properly written manuscript, based upon the historical nature of the psychological content, I strongly encourage authors to submit their manuscript in journals specializing in psychology, a few of those are listed below-
•    Journal of Personality and Social Psychology
•    Social Psychological and Personality Science
•    Journal of Family Psychology
•    Frontiers in Psychology
•    Journal of Personality
•    Psychological Science
•    Psychological Bulletin

Among other important concerns, the relevance of such study in modern and present societal context whether it is eastern or western culture, needs to be evaluated thoroughly. Study entirely based upon historically old documentation with limited and coarse contents cannot provide the detailed or ideal psychobiographic descriptions of an individual, something which authors have also realized and listed it as the major limitation. Furthermore, exploring the data of personality descriptions in a group of social elites does provide useful information, however, it certainly limits the applicability to general and targeted population for whom such psychological studies are needed.  
Additionally, authors should elaborate the relevance of such study in the modern context in the discussion and conclusion.

Author Response

Dear Dr. Renee Zhang: Thank you very much for sending us the email regarding our submission, and for offering us an opportunity to revise it.  Below we outline in detail about the changes we have made according to your and our Reviewers’ comments.  Our answers are in these big brackets ‘{}’.

To Assistant Editor, Dr. Zhang

AE1. Your manuscript has now been reviewed by experts in the field. Please strengthen the link with psychiatry during revisions, which is very important.

Answer: {Thank Dr. Zhang for the very important suggestion.  We have strengthened the link of this study with psychiatry, please see our adding in the Abstract (lines 15-17), Introduction (lines 99-101), and Discussion (lines 291-292; 301-302; 307-308).}

To Reviewer #1

R1.1. Authors present here an interesting work exploring the detailed associations between personality traits/facets and life outcomes with the progression and development of in a group of social elite from Imperial Chinese history.

Using revised NEO Personality Inventory (NEO-PI-R), and Destiny Evaluation Questionnaire (DEQ), authors analyzed the personality traits/ facets and factors of destiny between 18 characters, the Macrobian group, and 30 in the counterpart group and found that the less self-indulgence or self-focused attention, and more devotion to and interdependence between individuals were beneficial to several aspects of individual destiny in Imperial China.

Although I appreciate the authors attempt for such an interesting study with well-presented data and properly written manuscript, based upon the historical nature of the psychological content, I strongly encourage authors to submit their manuscript in journals specializing in psychology, a few of those are listed below- (Journal of Personality and Social Psychology, Social Psychological and Personality Science, Journal of Family Psychology, Frontiers in Psychology, Journal of Personality, Psychological Science, Psychological Bulletin)

Answer: {Thank our Reviewer #1 for the nice words and suggestions. As we have answered Dr. Zhang (Assistant Editor, AE1), we have highlighted the link between our study and psychiatry, especially personality disorders, please see Abstract (lines 15-17), Introduction (lines 99-101), and Discussion (lines 291-292; 301-302; 307-308).} 

R1.2. Among other important concerns, the relevance of such study in modern and present societal context whether it is eastern or western culture, needs to be evaluated thoroughly. Study entirely based upon historically old documentation with limited and coarse contents cannot provide the detailed or ideal psychobiographic descriptions of an individual, something which authors have also realized and listed it as the major limitation. Additionally, authors should elaborate the relevance of such study in the modern context in the discussion and conclusion.

Answer: {This is a very nice idea, thank our Reviewer #1. Accordingly, in the current draft, we have added statements about the relevance of such study in modern societal context whether it is the eastern or western culture in the last paragraph of Discussion (lines 298-300) and Conclusion (lines 307-308).}

R1.3. Furthermore, exploring the data of personality descriptions in a group of social elites does provide useful information, however, it certainly limits the applicability to general and targeted population for whom such psychological studies are needed.

Answer: {Thank our Reviewer #1 for the great notice. We have added one sentence over this point in our limitation part (lines 291-292) in the current draft.}

We hope our changes this time would satisfy you.  If we can be of further assistance, please feel free to contact us.

Yours,

Dr. Wei WANG, Department of Psychology, Norwegian University of Science and Technology. Trondheim, Norway, Tel: +47-46542194; Email: <wew@ntnu.no> or <wangmufan@msn.com>

Reviewer 2 Report

The manuscript entitled “Social elite in Imperial China: their destinies documented by literature and personality defined by contemporary five-factor model” is a potentially interesting contribution to the extant literature on personality differences. There are, however, several issues that need to be addressed before publication is warranted.

One of the most pressing issues is the lack of clarity in the authors’ writing. They are the experts. The audience may not know much about the individual differences the authors tackle. Clear definitions must be provided. If I were the authors, I would consider re-writing the entire manuscript to ensure clarity in content, adequate syntactic structure, and suitable word choices.

 For instance, the abstract is difficult to understand. What is exactly the methodology the authors have used? What is their justification for using the selected methodological approach? The following statement appears to entertain a circular argument that remains at its core unclear: “The historical records in Imperial China present life-long data of social elite, which is an optimal way to study the associations in China.” Why would this methodology be optimal? My modest advice is to rewrite the abstract by considering the following items: (a) the research question or purpose of the study, (b) the methodology that the authors adopted, (c) the main results of the study written in plain English, (d) application and utility of the results.

 The introduction does not make a clear-cut case for the rationale upon which the study rests. The hypotheses are difficult to guess from the authors' writing. A rationale for each hypothesis needs to be put forth along with its supporting evidence.

 The literature review is largely missing.

 The method section requires organization and clarity to ensure that a broad audience can understand the procedure used. Interested readers may want to replicate it.

The results section needs to be organized by the hypotheses that the authors have chosen to test.

The discussion section may benefit from a stronger association with the content of the introduction.

Author Response

Dear Dr. Renee Zhang: Thank you very much for sending us the email regarding our submission, and for offering us an opportunity to revise it.  Below we outline in detail about the changes we have made according to you and our Reviewers’ comments.  Our answers are in these big brackets ‘{}’.

To Assistant Editor, Dr. Zhang

AE1. Your manuscript has now been reviewed by experts in the field. Please strengthen the link with psychiatry during revisions, which is very important.

Answer: {Thank Dr. Zhang for the very important suggestion.  We have strengthened the link of this study with psychiatry, please see our adding in the Abstract (lines 15-17), Introduction (lines 99-101), and Discussion (lines 291-292; 301-302; 307-308).}

 To Reviewer #2

R2.1. The manuscript entitled “Social elite in Imperial China: their destinies documented by literature and personality defined by contemporary five-factor model” is a potentially interesting contribution to the extant literature on personality differences. There are, however, several issues that need to be addressed before publication is warranted.

One of the most pressing issues is the lack of clarity in the authors’ writing. They are the experts. The audience may not know much about the individual differences the authors tackle. Clear definitions must be provided. If I were the authors, I would consider re-writing the entire manuscript to ensure clarity in content, adequate syntactic structure, and suitable word choices.

Answer: {Thank our Reviewer #2 for pointing this issue out.  We do agree that making the audience fully understand this study is very important.  In this version, we have added a further definition of individual difference (i.e., personality) in the Introduction (please see lines 41-43).  Furthermore, we have asked the English editing service of MDPI to help us to tune the English language of our manuscript.} 

R2.2. For instance, the abstract is difficult to understand. What is exactly the methodology the authors have used? What is their justification for using the selected methodological approach? The following statement appears to entertain a circular argument that remains at its core unclear: The historical records in Imperial China present life-long data of social elite, which is an optimal way to study the associations in China. My modest advice is to rewrite the abstract by considering the following items: (a) the research question or purpose of the study, (b) the methodology that the authors adopted, (c) the main results of the study written in plain English, (d) application and utility of the results.

Answer: {Thank our Reviewer #2 for the notice regarding the Abstract, we have rewritten the Abstract part in the current version, by (a) illustrating the research question and purpose, (b) re-organizing methods, (c) facilitating results, and (d) extending conclusion (please see the changes in page 1).}

R2.3. The introduction does not make a clear-cut case for the rationale upon which the study rests. The hypotheses are difficult to guess from the authors' writing. A rationale for each hypothesis needs to be put forth along with its supporting evidence.

Answer: {Again thank our Reviewer #2 for the thoughtful comments. In the current draft-Introduction, we have input grit (like conscientiousness), passion, and mindset as supports of rationale (lines 66-71), more description of Chinese culture and psychiatric/ psychological world (lines 81-86), the relationship with personality development and psychiatry (lines 96-101), and the re-arrangement of hypotheses (lines 112-115), and we have re-articulate the whole layout of Introduction accordingly (rationales and hypotheses).}

R2.4. The literature review is largely missing.

Answer: {Following our Reviewer #2, we have added references to the Introduction part, regarding the individual differences/ personality (lines 41-43), grit (lines 66-71), and Chinese culture (lines 81-86). In this sense, we have added 10 more pieces of reference in total (please also see our reference list) this time.}

R2.5. The method section requires organization and clarity to ensure that a broad audience can understand the procedure used. Interested readers may want to replicate it.

Answer: {We are sorry for the unclear description of Methods section, especially 2.1. Characters studied in our previous draft. In the current version, we have added the dynasties and the selection criteria for the social elite characters (please see lines 122-126).}

R2.6. The results section needs to be organized by the hypotheses that the authors have chosen to test.

Answer: {Yes, thank our Reviewer #2. In the current draft, we have put two obvious subtitles (3.1. and 3.2.) into the Results section, corresponding to the hypotheses testing (please see lines 187 and 204).}

R2.7. The discussion section may benefit from a stronger association with the content of the introduction.

Answer: {Thank our Reviewer #2 for the insightful notice. Accordingly in the current draft, we have added the relations of grit and its brain structure development (272-275), the associations with personality disorders and contemporary society (lines 298-300, 307-308) in the Discussion section, which is mentioned in Introduction.}

We hope our changes this time would satisfy you.  If we can be of further assistance, please feel free to contact us.

Yours,

Dr. Wei WANG, Department of Psychology, Norwegian University of Science and Technology, Trondheim, Norway, Tel: +47-46542194; Email: <wew@ntnu.no> or <wangmufan@msn.com>

Round 2

Reviewer 1 Report

I really appreciate the authors for including the required changes in the manuscript, particularly in the discussion section highlighting the potential application of the current study to psychiatry in modern context.
I have no further comments or suggestions.

Author Response

Thank our Reviewer #1 for your approval and encouragement, Happy New Year 2023 from Wei Wang and coauthors!

Reviewer 2 Report

The article entitled “Social Elite in Imperial China: Their Destinies Documented by Literature and Personality Defined by Contemporary Five-Factor Model” has been adequately revised by the authors.  The information that has been added has introduced clarity to the literature review that the authors performed. In its current format, the study makes a substantial contribution to the literature on individual differences. Its unique viewpoint is one aspect of its content that makes it valuable. In my modest opinion, its publication is warranted. Of course, the discussion section may be further augmented with a broader review of the implications and applications of the results of the study. However, even in its current format, the discussion section is adequate. Additional proofreading may be needed though. For instance, the title seems to be missing an article and a pronoun. Consider the following changes: “Social Elite in Imperial China: Their Destinies as Documented by the Historical Literature and their Personality as Defined by the Contemporary Five-Factor Model”.

Author Response

To Reviewer #2 and Dr. Zhang (Assistant Editor)

First of all, happy new year 2023 to you. Thank you very much for your quick review of our revised manuscript.  Below we outline in detail the changes we have made according to our Reviewer #2’ comments.  Again, our answers are in these big brackets ‘{}’.

Q1. The article entitled “Social Elite in Imperial China: Their Destinies Documented by Literature and Personality Defined by Contemporary Five-Factor Model” has been adequately revised by the authors.  The information that has been added has introduced clarity to the literature review that the authors performed. In its current format, the study makes a substantial contribution to the literature on individual differences. Its unique viewpoint is one aspect of its content that makes it valuable. In my modest opinion, its publication is warranted.

Answer: {Thank our Reviewer #2 for your encouragement.}

Q2. Of course, the discussion section may be further augmented with a broader review of the implications and applications of the results of the study. However, even in its current format, the discussion section is adequate.

Answer: {It is very kind of you, our Reviewer #2, thanks. You have provided us an optional choice of “its current format, the discussion section is adequate”, therefore we keep our discussion (and conclusion) part unchanged.}

Q3. Additional proofreading may be needed though. For instance, the title seems to be missing an article and a pronoun. Consider the following changes: “Social Elite in Imperial China: Their Destinies as Documented by the Historical Literature and their Personality as Defined by the Contemporary Five-Factor Model”.

Answer: {The changes in the title, proposed by our Reviewer #2, are more propriate and clearer in meaning, thank you. We added these changes to our title (please see lines 2-4).}

Dear Reviewer #2 and Dr. Zhang, please feel free to contact us if we can be of further assistance.

Yours,

Dr. Wei WANG, Department of Psychology, Norwegian University of Science and Technology, Trondheim, Norway. Tel: +47-46542194, Email: <wew@ntnu.no> or <wangmufan@msn.com>